# Flexible Training Planning Coupled with Flexible Assessment: A 12-Week Randomized Feasibility Study in a Youth Female Volleyball Team

**DOI:** 10.3390/children10010029

**Published:** 2022-12-24

**Authors:** Manuel Loureiro, Isabel Mesquita, Ana Ramos, Patrícia Coutinho, João Ribeiro, Filipe Manuel Clemente, Fábio Yuzo Nakamura, José Afonso

**Affiliations:** 1Centre for Research, Education, Innovation, and Intervention in Sport (CIFI2D), Faculty of Sport of the University of Porto (FADEUP), Rua Dr. Plácido da Costa 91, 4200-450 Porto, Portugal; 2Football Department, Lusophone University of Porto, 4000-098 Porto, Portugal; 3Escola Superior Desporto e Lazer, Instituto Politécnico de Viana do Castelo, Rua Escola Industrial e Comercial de Nun’Álvares, 4900-347 Viana do Castelo, Portugal; 4Research Center in Sports Performance, Recreation, Innovation and Technology (SPRINT), 4960-320 Melgaço, Portugal; 5Instituto de Telecomunicações, Delegação da Covilhã, 1049-001 Lisboa, Portugal; 6Research Centre in Sports Sciences, Health Sciences and Human Development, CIDESD, University of Maia, ISMAI, Av. Carlos de Oliveira Campos, 4475-690 Maia, Portugal

**Keywords:** meta-assessment, bidirectional feedback, flexible planning, qualitative randomized study, volleyball

## Abstract

According to the Quality Education and Gender Equality ambitions established at the 2030 Agenda for Sustainable Development Goals, we aimed to test the feasibility of a flexible planning and assessment process, using ongoing, bidirectional feedback between planning and assessment. Eighteen players (11.5 ± 0.5 years of age) from a U13 female volleyball team were randomized into an experimental group (in which the plan could be changed daily) or a contrast group (pre-defined planning, adjusted monthly). The pedagogical intervention lasted three months. Besides ongoing daily assessments from the training practices, the Game Performance Assessment Instrument was adopted as a starting point for the weekly assessments in 4 vs. 4 game-forms (i.e., the instrument was modified monthly based on feedback from the training process). Information from daily and weekly formal assessment was used in the planning of the experimental group, and monthly in the contrast group. Data suggested that pre-established and strict planning (even updated monthly) failed to fit current learner needs. Over 12 weeks, the pre-established planning suffered regular modifications in the experimental group, and the assessment tool changed monthly. In conclusion, both planning and assessment should be open and flexible to exchange information mutually, and support the design of tailor-made learning environments.

## 1. Introduction

Quality in Education and Gender Equality are two core items (goals 4 and 5, respectively) of the 2030 Agenda for Sustainable Development Goals (SDGs). The endless search for better learning strategies (point 4), framed upon learner-centered approaches [1,2], and the breaking away from the common application of investigation protocols in male samples (point 5)—which is typical of Sports Sciences [3,4]—are pinpoints of this work. Learning depends on inner transformations, producing complex, interwoven interactions between educational environments and intra and interindividual differences in response to dynamic, evolving paths (i.e., bidirectional stimuli and interactions) [5,6]. Consequently, predicting the results of any learning process, in terms of its magnitude, quality and timings, is a very difficult—perhaps impossible—task [7,8]. The non-linear nature of educational settings entails that identical inputs may produce vastly different outputs, whereas other inputs may result in the same output [9,10]. *Actual* learning should therefore be deeply intertwined with planning and assessment, but these are often essentially predetermined and/or standardized [11,12,13].

Planning the learning process is thoroughly recommended, but the level of detail and time ahead might be questionable [13,14]. In contradiction with non-linear premises, teaching and training processes tend to over-emphasize pre-planning, including pre-established contents and their sequences, as well as time restrictions based on the idea that each training phase will enhance the subsequent one in a predictable manner (i.e., periodized programs) [14,15]. In sports training, coaches often use periodized programs, which establish the sequence and timing of events [8,16]. This kind of approach could force coaches to follow pre-planned content, ignoring important incoming information from the learners. For instance, in team sports, Gavanda et al. [17] aimed to compare different resistance training programs (all of them pre-established) in American football and did not find any benefit of one to the others, which shows the inability to predict the players’ response to practice. The motivation behind this study is based on the idea that there is no optimal sequence to learning skills [18]. Therefore, the best way to drive the planning is by constantly assessing and screening the unfolding of the actual learning process, detecting fragilities and opportunities to act over time [8,13].

For responding to non-linear features of education and learning [9,19], perhaps there should not exist strict start and end points to each block of content; moreover, the timing of each content section should be established individually [8,20]. Furthermore, flexible plans, coupled with ongoing flexible assessments, may respond better to the non-linearity of learning, keeping the focus of the process on the learner instead of the planning [21,22]. 

Assessment tools should be applied continuously and incorporate flexible features [23,24]. In this vein, assessments should occur with a certain regularity (preferably ongoing, i.e., “each exercise is a test”) to create a more accurate picture of the *real* improvement on learning. Complementary to this, being flexible enough to adapt to different moments and needs of the learners (i.e., move away from standardization and become an evolving process) is important [11,25]. In this regard, recently, Atkinson and Brunsdon [24] applied a flexible approach to assess students in Physical Education classes that adapt better to their needs, especially in soccer, basketball and volleyball underlying that assessment in sports should go beyond measuring performance. However, currently most of the assessment tools used in Physical Education and sports training, as EUROFIT of FMS, regulate their assessment based on average values (abstract, and away from the real context) acting like a “one-size fits all”, and ignoring the individuality of the learner and the context [26,27,28].

Evaluating the assessment tools and reflecting on their utility in planning is called “meta-assessment”, and it is fundamental to keep an open path between planning and assessment [25,29]. A straight connection between planning and assessment implies a bidirectional exchange of information [11,13]. So, planning should constantly inform the assessment about the main goals and execution criteria to fulfill [11]. Concomitantly, this information might result in a deeper reflection on assessment: whether the criteria are adjusted to the learners, and whether the assessed content is related to established planning goals [24]. Doing so, sport practitioners are working on building appropriate learning environments. Appropriateness emerges as a concept whose application varies depending on the individuals and the context in which they operate, and build their sporting identity [30].

For studying bidirectional links between planning and assessment, volleyball portrays an adequate application field given its technical rigor, as well as dynamic and complex action possibilities among players [31,32]. Studies focused on pedagogical interventions in the context of volleyball are usually performed with pre-defined planning. To exemplify, Sgrò, et al. [33] aimed to examine the effect of a determined instruction plan in volleyball learning; however, during the 13-week period, the plan was not adjusted based on the learner’s evolution. This largely contrasts with our idea of using the assessment as a source of information to correct planning. The same occurs with the assessment tools used in those studies, which are the same at the beginning and end of the pedagogical intervention [33,34]. Besides that, the application of randomized interventions in sports training and scholarly contexts might bring some implementation issues, especially when those applications imply the whole training time, and are across every training day. This randomization process implies the total division into different groups that do not mix in any part of the learning process. 

This study aimed to test the feasibility of flexible planning coupled with a flexible assessment process in a longitudinal experimental trial (12-weeks) with parallel randomization, launching ongoing, bidirectional feedback between plan and assessment of learning—especially of the dynamic, ongoing process of assessing and readjusting the planning in a young female volleyball team (i.e., real-life context). Overall, this project presents the novelty of an evolving planning and assessment process, due to its strong, ongoing, and bidirectional links, enhancing a process-focus instead of a traditional product-focus design. Furthermore, to guarantee that the effects in learning were not merely due to time effects, the longitudinal nature of the intervention, along with the implementation of two contrasting randomized groups ensured a greater quality of data interpretation.

## 2. Material and Methods

### 2.1. Design

We adopted the CONSORT guidelines [35]. The purpose was to implement longitudinal experimental research with parallel randomization that effectively assessed qualitative and quantitative learning changes when comparing two approaches to planning. All the data were collected in the training facilities of the club, throughout 12-weeks in a total of 34 training sessions. The study was conducted in accordance with the Declaration of Helsinki and approved by the Ethics Committee of the Faculty of Sport of the University of Porto (CEFADE 30 2021), and financed by the Portuguese Foundation for Science and Technology (reference: EXPL/CED-EDG/0246/2021). The original protocol is available in the English language upon reasonable request.

### 2.2. Participants

The sample was eligible by convenience [36], consisting of a female mini-volleyball team (n = 18, age 11.5 ± 0.5) in one of the most prestigious Portuguese volleyball clubs. These players were considered ‘information-rich’ due to being at the beginning of their sporting pathway, so they did not have tactical or technical preconceptions on volleyball practice and were actively engaged in participation. The club has more than 100 national titles in volleyball, participation in European competitions, and is known for its excellence in younger teams. It also has top training facilities which largely helped the practice conditions, and consequently the data collection. The previous coaching staff had divided 18 players into two teams: A (more advanced) and B (less advanced), according to their skill level. Any player could be excluded from the investigation if, at any moment, they left the club or withdrew from the practice of volleyball. During the 12-week period, the coaching staff were able to deliberate whether any other player of that age should be integrated into this investigation. In this case, the referred player should be randomized into one of the groups. The team was composed of 10 players deemed of skill level A (most advanced, 11.3 ± 0.3 years of age) and 8 players deemed of skill level B (less advanced—11.9 ± 0.3 years). Skill level was determined here in a comparative manner, and skill level A did not necessarily mean a high absolute skill level. Upon randomization, the experimental group was composed of 9 athletes (5 from level A and 4 from level B; 11.6 ± 0.5 years) and the contrast group of 9 athletes (4 from level A and 5 from level B; 11.3 ± 0.5 years). Upon beginning this research, whereby two of the authors became the coaches of this team, the players from both teams were randomized into two separated groups of 9 players (i.e., using an allocation ratio of 1:1) through the tool http://random.org/sequences accessed on 30 March 2022. The randomization was performed by a researcher not engaged with the interventions, and unknown to who the athletes were. To further ensure concealment of allocation sequence, the main researcher (who also implemented the interventions) only had access to the groups on the first day of the intervention. The first author plays a double role in this investigation as head coach and researcher [37]. The players and their parents were fully informed on the research goals and signed an informed consent form that had been previously approved by the Ethics Committee. Anonymity was ensured (e.g., not using the proper names of players) and players were also informed of the possibility of withdrawing their participation at any moment. Figure 1 synthesizes the study design.

### 2.3. Interventions

Upon an initial diagnostic evaluation of the players’ skills during the first week of practice, a periodized program was designed as described in Appendix A (Periodized planning), with a weekly formal assessment moment every Saturday, as it was the last training session of the week. This periodized program focused largely on tactical and technical contents (i.e., on learning the specific volleyball actions) and not on physical or psychological factors. Both groups were guided by the same Game Model (GM). The GM established technical and tactical principles of play, which are fundamental for team organization [38,39]. Thereby, those principles largely informed the initial planning. Specifically, the creation of the GM was conducted according to the technical and tactical skills of players, their past experience in the sport, and the club’s philosophy. This philosophy is based on a long-term sports development, preparing the youth players to integrate into the adult team, which has competed in the highest Portuguese division for decades. This investigation occurred during the last 3 months of the season; therefore, the previous coaching team that had a reasonable knowledge of the players’ skills and team dynamics, established the GM. The GM required some advanced control of the basic volleyball actions (e.g., set, pass, and spike) as well as a reasonable level of game-related knowledge (e.g., exploring different attacking zones and tempos). The attacking zone is determined by where the attacker contacts the ball according to the zones of the volleyball court, while attacking tempo is defined by the relative timing between the attacker and the setter’s actions [40].

The experimental group followed an open learning process with regular interchanging information between planning and assessment, whereby the planning could be altered daily. Thus, the planning process tried to satisfy the players’ needs in different moments, and the adaptations performed to the initial planning were based on ongoing assessments (i.e., the coaches’ daily perceptions of how the players were evolving and adapting to the training process, including daily field notes). Weekly discussions between the head coach/main researchers and the research team complemented the planning process. Conversely, the contrast group followed a pre-planned, periodized intervention, which could only be adapted monthly. Therefore, both groups could deviate considerably from the original periodized plan, although those deviations would occur at different time points. Importantly, although the plans had to be followed for at least 1 month in the contrast group (i.e., the contents and their specificities had to be respected for that period), the specific exercises and their sequences could be planned daily to better deliver a proper pedagogical process.

### 2.4. Assessment

We emphasized the idea of turning every practice into an assessment moment, where the coach was responsible for taking notes on the performance of players, planning limitations during the learning process, and thoughts on how players were reacting to the current planning. This ongoing assessment provided ongoing insights that were used to understand the players’ evolution as well as individual needs and helped the coach and research team to reflect about the learning process. Therefore, ongoing bidirectional links were established between planning and assessment. All of the information collected during the daily practices (through reflections), was complemented by punctual formal assessments that were held in every last session of the training week (Saturday), during 10 min of 4 vs. 4 games between both groups. The games were recorded by an HD camera in the back of the court and at around 2 m height to have a clear vision of both sides of the court. Here, the baseline assessment tool was the Game Performance Assessment Instrument (GPAI) [41], as it delivers a holistic overview of how the learning process is translating into meaningful behaviors in game-like situations. The starting version of the GPAI was the one used in the INEX study [42]. In this specific study, the GPAI was divided into four major analyses: efficiency; efficacy, adjustments, and decision-making. In each one of those categories, every contact of the player with the ball was considered as “appropriate” or “inappropriate”, depending on whether they fulfilled the pre-established criteria or not. During the day, after the formal assessment (i.e., on Sunday), the data collected by the video was analyzed (every time by the same person belonging both to the research and coaching teams) in a qualitative and quantitative fashion. All the actions were analyzed with a qualitative approach, according to GPAI criteria. Then, GPAI turned the “appropriate” and “inappropriate” actions into quantitative indexes of efficacy, efficiency, performance, and decision-making.

Both the GPAI outcomes and the daily reflections derived from the training process were discussed between the research team and coaches each Monday to plan the following practices of the experimental group, and for outlining potential necessities for the contrast group (which could only be implemented after one month). Once a month, a special meeting between the research team and the coaching staff was held to debate the developments of the training process within the contrast group and determine changes to the planning. Moreover, the starting version on GPAI was changed every month both for its categories and respective criteria (i.e., meta-assessment process), according to coaching feedback of their specific concerns, as well as players’ evolution in game-related knowledge. Doing so, we aimed to fit the assessment tool as much as possible to players’ needs in every moment.

Considering the goals and nature of this randomized study, the outcomes are mostly qualitative, attempting to provide an account of the evolution of this process of bidirectional links between planning and assessment. Since meta-assessment was employed, not even the original GPAI can be directly compared to the endpoint GPAI, as the instrument was strongly modified during this study (as this was one of the goals established by the research team and it is coherent with a flexible approach to planning). This is possible to verify in Appendix A.

### 2.5. Blinding

Due to the ethical concerns involved with this study, both those implementing the interventions and those participating in them (i.e., the players) were fully aware of the goals of this trial, although the players were not aware if they had been incorporated into the experimental or contrast group. The main author was not blinded to the interventions, but the remainder of the research team was, and the weekly meetings (and the changes emerging from those meetings) were decided with most of the research team remaining blinded to the participants in each intervention (apart from the main author).

### 2.6. Data Analysis

Data analysis can be divided into three major aspects: (i) descriptive and ongoing report of a player’s evolution through the daily performance in practice; (ii) reflection about the learning-process, its implications on the assessment tool, and the barriers and opportunities on the randomized intervention; (iii) analysis of GPAI data—more important than the comparison of performance between different moments, the interpretation of data to guide subsequent planning, and adaptation of the assessment tool in research-coaching meetings.

## 3. Results

All 18 players finished the season, and thereby 100% of the eligible sample was recruited and completed the experiment (Figure 2). The focus of the results was not on the individual evolution of the players, but the adaptations to the planning performed in both groups, and the readjustments performed throughout the assessment moments. The analyses were focused on the training process, namely the interplay between planning and assessment in both groups, and the barriers and opportunities associated with this type of research.

### 3.1. Adaptation of Planning in Light of Ongoing Assessments

*The starting point* for planning and determination of drills in training and the pedagogical strategies used were deeply connected with the players’ skill level, and consequently GM premises. The first assessment moments (i.e., the coach’s perceptions during the 1st and 2nd training weeks) suggested that the initial planning required adjustments. At this point, a low understanding of the game (high values of inappropriate actions in tactical domains, such as spike and upper-hand/tennis serve), and consequently a weak interpretation and execution of the game, were noted in both groups. In this way, we noticed that the GM was, in part, misadjusted from reality, and required adjustments. Ethically though, no decisive conclusion should be stated at the moment: (i) on the one hand, the experimental group may seem to benefit from more agile changes to the planning that are better adjusted to their current state of development; (ii) on the other hand, it is possible that a more stable plan could provide the necessary stability, and that the contrast could benefit from persisting for an extended period before changing the overall plan.

To exemplify, we felt the necessity to spend much more time practicing (i.e., was the central topic of the week for 12 weeks in the experimental group, and only in 8 weeks of the contrast group) basic ideas of passing and setting to create better conditions to spike. We implemented those adaptations in the experimental group, while the contrast group spent some time working on the spike technique and approach. Despite being subjected to the pre-planned contents, the contrast group benefited from adjustments in the exercises, which were progressively more tailor-made to help the players in the contrast group to evolve.

We thought that it could be possible to reach more advanced volleyball actions such as (i) jump-float serve; (ii) different attack tempos; (iii) jump-set; (iv) variations in attack in zone 3, as they were part of the GM. However, during the initial weeks, we noticed that the experimental group should follow a strongly different progression, specifically requiring a less ambitious plan (i.e., more strongly focused on some basic game actions, such as passing, setting, and displacement), especially from the 3rd week of intervention onward (Appendix A). Those differences were based on daily field notes, and useful information from the GPAI assessments. After the first month of the intervention (i.e., in May), some of the initial goals were readjusted in the contrast group, such as different attacking tempos that were withdrawn from the planning. At the end of the second month, we noticed that a few pre-established contents were too advanced for the real state of the performance of the whole group, such as jumping set or variations in attack on zone 3. During the last month of the intervention, the major difference between groups was the introduction (as pre-established) of the jump-serve and the introduction to blocking ideas that were also not used in the experimental group.

*Ongoing feedback between planning and assessment* was the significant input of this investigation into the learning process. For example, the utility of exploring the attack on zone 3 in May, or even the approach to attack, and how to attack a high ball all in the same month. Near the beginning of the second month, this point turned into a growth opportunity, as a coach. Sometimes we felt the content planned for the contrast group was not the most adjusted and still created stimulating tasks for the learner, based on that pre-established content. During the month of June, we tried to introduce the block, and automatically felt that this was unimportant entirely. 

### 3.2. The Contribution of the More Formal Weekly Assessments

During the process, the GPAI was considerably modified to fit the players’ necessities and the coach’s concerns (e.g., to assess the set action in different contexts such as passing and setting; change service assessment to only one category—upper-hand service). Those changes occurred in efficacy measures (e.g., removing punctuation from passing and digging), efficiency (e.g., adding different weights to continuing attacks), decision-making (e.g., when attacking instead of attacking to the right spot), and adjustment criteria (e.g., reading and reacting to a Freeball situation—i.e., occurs when the opponents predictably send an easy ball, thus not having the possibility of attack [32]). All of those adjustments will be developed further. Accordingly, from the first GPAI version to the last one, almost all the criteria were modified and adjusted considering the analysis of coaching staff and the players’ needs—see examples in Appendix A.

To exemplify, we excluded the “underhand and jump serve” and combined all the observation grids for the “overhead serve”. In the first GPAI modifications (version 2) we made a few adjustments in the technical criteria of setting and passing; however, in the second moment we felt the need to change the criteria slightly: the set as a volleyball action is different when is used as first contact (i.e., dig or receive), or as the second contact (i.e., setting). For this reason, we split the set assessment into two categories: the first contact, and the second contact with different criteria to assess the “same action”. In addition, the need for understanding whether the players read and react correctly to a freeball situation emerged because that game phase has specific characteristics. Therefore, a category of ‘freeball’ was created to analyze if the setter could run this type of transition, and if the defenders adopted the correct behavior (i.e., an appropriate occupation of the pitch).

We adjusted some criteria about spike and serve actions (GPAI version2), since we felt that the players were so focused on changing and improving technical issues that they neglected the tactical issues (i.e., to where should I serve or attack?). So, in GPAI version 2, we removed all the decision-making categories to attack and serve. Later, in the third GPAI, we returned to analyze the decision-making in attacking, but we completely changed assessment lenses. Instead of looking at whether the player attacks in the right spot, we assessed whether, in the right conditions, the player chooses to spike. Doing so, we had a different viewpoint of the attack once we were concerned about the player being capable of spiking in the right spot (i.e., making the best decision), as well as controlling the technique and knowing how and when to use it. The blocking action follows, in part, the spike assessment. Once that occurs, we start assessing the technique demands and finally assessing whether the player decided correctly when they should block or not.

Evaluations of actions’ efficacy were adapted over time from the first version of GPAI that only contemplates three categories: (i) error; (ii) continuation; and (iii) point. Additionally, all the game fundamentals (i.e., service; reception; set; spike and block) were considered in the same way, even when they have considerable contextual differences. Thus, we decided to remove the point criteria of continuous actions, and adjust the terminal actions based on Palao et al. [43] by creating different weights according to the difficulties imposed on the opponent.

*The meta-assessment or assessing the assessment* occurred during the monthly meetings between the coaching and research team. Those adaptations were consequence of reflective moments on the planning, and how the assessment tool could be adapted to accomplish learning goals. In terms of efficiency, the initial version of GPAI had 23 variables within different volleyball actions; in contrast, the final version had only 13 variables focused on the same volleyball actions—see Appendix A. Moreover, to keep the data collection simple and precise, the “adjustments” that contemplate the tactical behavior of the player in the game context also decrease, from 13 to 11 variables. 

### 3.3. Barriers and Opportunities: On the Feasibility of Implementing This Research Design

With the *randomization* into two groups with different planning methodologies, the coaching staff had to plan two different practices, and at the same time be able to guide different exercises with their specificities and goals. To correctly apply this process, we needed a larger coaching staff. For instance, several players that started playing volleyball to enjoy more time with friends were set up to practice in separate groups; this could, in part, influence their motivation to practice, as it was verbally expressed by a few players in the beginning of the pedagogical intervention. To minimize the damage of that split, in the last training before competition, part of the practice participants were divided into teams (as they will play in the tournament) and not into groups (experimental and contrast); this time, around 30 min in one sporadically organized session did not represent a bias in the training division. 

*A mixed-level group* (i.e., players from team A and players from team B performed together) started as a problem, especially regarding the motivational issues of the most advanced players. However, it became helpful for several reasons: (i) the building of a more extended and cohesive group of players, that did not practice only in small teams, which became visible in tournaments with players of different groups supporting each other; (ii) players practiced with higher-skilled players and with lower skilled players, turning the whole group into more extended and homogeneous one which, even if it not the most essential thing in the short term, could be a significant input for the club in a medium-long term; (iii) all these features contributed to a more diverse environment for practice, as playing with different athletes exposed each player to different constraints and promoted different strategies.

*The adherence rate of players* was similar in the experimental groups (mean: 78%, minimum: 60%, and maximum: 90%) and contrast group (mean: 81%, minimum 65%, and maximum: 94%) consistent with a realistic training process, especially with such young players who were not yet engaged in weekly formal competitions. Non-adherence to practice can cause multiple issues, such as a lot of time between training stimuli, and missing learning moments of specific content. 

Many *external constraints* have also affected the application of pre-established planning. As an example of such external constraints, we had to deal with the inhibition of using the training hall to practice due to other competitions; on two consecutive Mondays, the group was forced to cancel the practice, and it became impossible to develop the pre-established planning in the contrast group. In that case, we kept the planning and went through the missing session. In this vein, other moments forced the coaches to develop the training in different training gyms, with extra space, number of balls or number of courts available. 

### 3.4. Endpoints: Convergences and Divergences between the Two Paths

Besides monthly adaptations in the planning of the contrast group, at the end of the intervention, there were a few relevant differences between the planning of both groups such as introduction to jump-float serve, blocking and jump-setting in the contrast group, which the experimental one did not follow. Ongoing assessment, especially through daily reflections after practice, resulted in a simpler planning with essential actions in the experimental group, while the pre-established planning (i.e., contrast group) was more ambitious. Moreover, this discrepancy meant that the experimental and contrast groups followed (at least in part) different paths during the learning process. However, the monthly adjustments in the pre-established planning (i.e., contrast group) considerably contributed to reducing the gap in the differences between planning for the experimental and contrast groups. 

Most of the athletes ended the 12-week intervention more proficient than they were in the beginning. However, the essential question at this point was “what does the coach/teacher think that is more important to this specific group of learners?”. In our intervention it was clear that, based on the club’s philosophy, to improve passing and setting one should guarantee sustainable interpretation of GM, and only with the control of those actions, should the players be able to interpret and use correctly most advanced actions (such as the jump-set or spike). 

Both experimental and contrast groups had carefully planned practices, and players improved their overall performance in the game. In competition, although the practice group was different from the competition one, we noticed a stronger spirit of mutual help between the whole group of players, which it is a considerable positive point in this process. Here, the most important outcome is that the whole group (i.e., both experimental and contrast) improved in the interpretation of GM, and in execution of volleyball actions. In this sense, even though the experimental and contrast groups followed different paths, especially in planning contents, it is important to notice that both improved their performance in official competitions.

## 4. Discussion

The present study explored the feasibility of a randomized investigation in a youth female volleyball team, namely how a flexible planning interplay with an ongoing and adaptable assessment, could cooperate to improve learning [11,13]. The results showed that the bidirectional feedback between planning and assessment is feasible, as well as this exchange of information resulting in ongoing changes in planning and in the way as players were assessed [12,21]. Accordingly, the weekly assessment was revealed to be important, not only to determine the qualitative and quantitative aspects of a player’s performance, but also to enable coaches to devise appropriate learning contexts according to players’ needs (i.e., by constantly informing plans on possible readjustments). Thereby, one of the main results of the study was the adaptation made to the assessment tool over time (i.e., meta-assessment), and according to the development of players. Moreover, our data demonstrated that all the pedagogical interactions between coach and player can turn into informal assessment moments and provide useful insights into following learning tasks. The assessment tool, in this specific case the GPAI, became of paramount importance, especially regarding its flexible features. 

### 4.1. The Need for a Flexible Approach to Planning

Planning in a flexible fashion opens the door to adapt not only according to external issues (i.e., training facility unavailability) [44], but also by considering internal processes (i.e., learner’s development) as it was recognized in this study [45,46]. After the implementation of this investigation, it became clear that there are multiple factors influencing the planning of the learning process that are out of control of the teacher/coach (e.g., external constraints as training center issues, adherence of players to practice) [47]. As it is possible to notice in our study, the comparison of content between the groups’ planning rarely matched. From the first moments, after a period of diagnosed assessment due to a specific deficit of game knowledge, coupled with some difficulties in performing many technical actions, led the coaching staff to re-think the following planning. This means that even “just” one month ahead, it was not possible to predict players’ evolution [48]. This finding is in sharp contrast with the study of Pliauga, et al. [49], where it is assumed that, in long-term planning, coaches/teachers know a priori the outcome promoted by different training approaches. Conversely, our findings strongly corroborate the assumptions of Niemi [50], namely that the non-linear nature of learning makes the outcome promoted by a training session unpredictable, and enhances the need for continuous adaptations to learners’ needs over time. 

So, it is urgent to change the paradigm of planning from pre-established content and timing to a flexible approach [8,51]. In this investigation, both groups received largely different planning, even after a periodical adaptation of the contrast group. There were other issues identified and reflected on by the coaching staff during the process, sometimes thinking how fair it is to expose a group to a practice exercise that is not the most appropriate (i.e., contrast group). Even so, independently of the content, improving a specific technique and using an action to increase game-related knowledge was possible. The essential important feeling, so far, was to show all the players that the experimental group was not “more important” than the contrast group. Thus, following an appropriateness-based intervention, the coach adapted each skill so that players could feel that every exercise of every group was useful to help them become better players.

### 4.2. The Need to Rethink Assessment

Across the pedagogical intervention, we observed that a substantial part of the assessment was conducted in a daily routine, through analysis of training and response of learners to practice [52]. Complementary to this, GPAI punctual and formal assessment added more information and some precise data to ongoing assessment, enhancing the learning process [41,53]. The first version of GPAI, broadly used in the literature was, in part, decontextualized. Underpinned with general technical ideas, some technical criteria were so generic that they did not provide helpful insight to the coaching staff. For example, in the first version of GPAI we assessed whether the player gets ready to attack after a dig. However, such an action might not be possible when, for instance, the setter digs the ball. Thus, our data showed the importance of using a representative assessment tool (i.e., contextualized to practice) that evaluates the learners’ performance and evolution accurately. Such an assumption contradicts the other studies that applied a single unmodified version of GPAI. For instance, studies used GPAI to compare technical and tactical behaviors with different court sizes in small-sided games [53]. In this way, the same criteria were used to different constraints (i.e., court size). In our investigation, we created a deep connection between planning and the assessment, raising the question of how we could build assessment tools that were constantly adapted over different learning moments [13,24]. 

Assessment should be applied as a regulator of the learning process and a collection of valuable insights that might be used to adapt subsequent teaching procedures. For example, our change of assessing the “set” and contextual interference in the same volleyball action—as they have other technical demands, such as the point of contact with the ball (higher is setting) or the use of the legs in the action (more critical in dig or reception). Therefore, during the investigation, we created constant updated versions of GPAI, more precise and capable of aiding the coaching staff in a more direct, quicker, and easier fashion—all versions in Appendix A. However, traditionally, assessment occurs in a formal way and at very specific time points during a learning process [54]. A clear example of that is the scholar context where assessment normally takes place at the end of a module, or period to grade students (i.e., summative assessment) [55]. In this way, the content is the center of the learning process and not the learner, as we argue [25].

So, assessment should be much more than just a formality, but a pedagogical tool that ongoingly informs the educational praxis as is suggested by Dixson and Worrell [56] or Leenknecht, et al. [57]. Additionally, the forms of assessment applied might be more variable: not limited to formal assessments such as exams or tests, but also promoting moments of self or peer-assessment, and consequently engaging learners in the learning process [58,59]. All the data collected were both in the formal assessment (i.e., GPAI) and ongoing informal assessment (i.e., daily practice, coaches’ notes) which were used to adjust the planning and design of learning environments of the experimental group during training sessions. Formative assessment only occurs when the outcome of the assessment is truly used in other learning activities [12,60]. It is critical to change the mindset of how we look at assessment, by putting the learner in the center of the process and thinking about how it can cooperate with learning [22].

#### 4.2.1. The Need for an Ongoing, Flexible Assessment

The main goal of assessment should not be to take different pictures, such as a snapshot of individual learning moments, but to create a more global view, or movie, of the process that allows for a more in-depth understanding regarding the evolution of the learner over time and consequently, to support the learning process [30,61]. We adapted the GPAI formulas for the efficacy and efficiency formula considering three actions for scoring points in volleyball (service, spike, and block), and actions that usually do not score points (passing, setting, and digging). By passing and setting, we can increase the possibility of scoring, but these actions usually do not allow for scoring a point [62]. In our viewpoint, this modification allows the coach and the player to more deeply understand the consequences of their own attack on the opposite team. Therefore, if we want the planning to be as flexible and moldable as possible to learners’ needs, coaches and teachers should be able to detect those needs [8].

Ongoing assessment seems to be, so far, the best way to see deficits in the learning process, as well as to be capable of measuring the learners’ evolution [63,64]. If the learner has other issues to solve over time, the assessment used should be able to detect various problems [30]. To deal with such complexity, assessment tools should acquire a flexible characteristic (i.e., being moldable, capable of measuring different variables, and changing the criteria) [24]. Flexible assessment tools ensure a clear overview of learners’ performance depending on their own needs [24,65]. To complement that, an ongoing use of those assessments allows the coach/teacher to detect different fragilities, and plan accordingly [66]. For example, suppose the significant difficulties an athlete, when learning to set in volleyball, will probably have with issues in the hand positions or reading the ball’s trajectory. Though, after a training period (to some players it might be a week, to others it might be a month or more), the challenges that players will face might be completely different, such as attacking player tempo or blocking opposition.

#### 4.2.2. The Role of More Punctual, Formal Assessments

Ongoing assessments during the week provide the coach with a considerable amount of data and valuable insights into the following week’s planning—as are daily reflection on the process and the response of learners to the practice [67]. That importance of reflection towards better planning strategies, and consequently better practices, is referred to by Downham and Cushion [68] while interviewing high-level coaches. In addition, ongoing assessment during the practices was always restricted to content used in those practices (i.e., if the practice does not explore the spike action for example, during the week we cannot understand the true needs of the learners in that action). To complement this idea, a formal weekly assessment allowed the coach to have a more precise overview about the performance of the players in the real game scenario (i.e., it is actually representative of teaching-learning context). 

Besides the crucial importance of the ongoing assessment of the learning process [69,70], a punctual moment could have important pedagogical implications [71,72]. In a certain way, in daily practice we can only extract information about the content that we practice; for instance, if I focus on the service action, I will have more data and conclusions about that specific moment. When we create an open assessment moment, even if the tool’s design had criteria, we allow the learner to express their performance with much more “freedom,” allowing us to detect different fragilities from daily practice. Those moments could act like motivational benchmarks in the learners’ process. In our study, weekly assessments of 4 vs. 4 games became one of the most desired moments of the week for many reasons: players want to compete, and essentially, the evaluation was constituted by something they enjoyed (i.e., playing volleyball). If on one hand, Shepard, Penuel, and Pellegrino [60] incentivize the formative assessment instead of using punctual assessment and grading to motivate students, on the other hand, both processes can complement each other. 

#### 4.2.3. Meta-Assessment as a Pedagogical Imperative

The meta-assessment, as an adaptation of the assessment tool to learners’ needs should be imperative to every learning process [71]. Staying away from the conventional idea of a pre-established tool with defined contents, tasks, and criteria, it seems to be time to move into more flexible and adaptable assessment tools. In this intervention, meta-assessment resulted in three structural modifications to the GPAI assessment tool. Those processes of meta-assessment might fit with different subjects addressed during a determined period, and constantly adapted to the group and the moment. This premise strongly supports the inter and intra-variability inherent to any learning process, which implies that not all learners should follow identical sequences and timing of educational contents, as advocated by van de Ruit and Grey [46] and Raviv, et al. [73].

We experienced the monthly special meeting between researchers and coaches as a rich moment of reflection, sharing and thinking about how we could extract useful information from the formal assessment. At the same time, those meetings not only focused on the assessment process, but also on how we manage the planning according to the new goals. In this vein, the assessment should not be the same for all the groups or even, to all the learners; it should be capable of being adapted according to the needs of the specific group of athletes, or a class [56,74]. So, this process of adaptation and changing the assessment mechanism implies deep reflection on evaluating one’s own assessment tool [29,75]. First, the constant adaptation of the tool implied a constant adaptation of the observer. Therefore, the first assessment, after changing GPAI, took more time due to adapting to new criteria and categories. Although, it was an excellent moment of reflection about what the coaches want from the players, how they want them to do each action, and what the following steps are in learning volleyball, coaches and teachers should often ask themselves: “is the assessment giving me useful insights?”; “is the assessment related to the actual planning?”; “what changes could be made in assessment to better respond to my worries?”; and beyond that, ”is the assessment responding to players worries and needs?”. Concluding, the adaptations in the categories of GPAI allow us to have a more precise view of the game and players’ needs; however, the constant adaption also had a few issues. We proceed to this adjustment because all of the players completely dominated “underhand service”, so it is an action that does not challenge the player. Moreover, it is not a technique used in most advanced levels of the game (i.e., from the beginning 6 × 6 to the senior levels). Consequently, it will not give any useful insight into the subsequent planning; on the opposite side, the jump serve is an action that the players cannot use in official competitions and, most importantly, none of them can execute. Figure 3 synthesizes the workflow of implementation, assessment, and planning.

### 4.3. The Need for Ongoing Feedback between Planning and Assessment

The initial planning was designed after an informal assessment that intended to diagnose the players’ abilities, and conversation with prior coaching staff about their level and the necessities of the team. The planning should be conducted by creating strategies in order to fulfill major difficulties of learners, and take them to the next level of performance [22,76]. Diagnostic assessment, commonly used in school and sports training, can be a useful tool to direct the first planning period (a session or a week, for example) [55,66]. The diagnostic assessment should be able to show what the global level of the learner in a certain activity are (i.e., what are the major difficulties of the students in a 4 vs. 4 volleyball game?). After that first assessment moment, all the planning gets a basic intent: to get better and improve in those difficulties [77,78]. Planning and assessment should cooperate in interdependence, responding to non-linear features of the learning process [13,79]. In this way, we acted differently from the majority of longitudinal experimental studies where there is no exchange of information between assessment and planning. As an example, Leonardi, et al. [80] examined the changes in tactical performance of female basketball players during a 4-month period, but without ongoing feedback, only focusing on the final product. Assessment moments might measure the evolution of learning, and constantly inform what should be the priorities in the subsequent planning [11,81]. During the 12-week process, we experienced this deep connection between planning and assessment. Over the time, the coach took notes about the difficulties of players, and some specific worries that should be analyzed by the assessment tool. At the same time, in the beginning of each week, in the research-coaching meeting we discussed the players’ performance during the assessment, as well as coaches’ feelings during the week and how those could be turned into useful information for planning. Maybe if there were specific issues detected, those might be used in subsequent planning and be the focus of the next assessment. For instance, what specific issues of passing were addressed in the training session? The assessment should focus on having a true idea of learned aspects and major difficulties. Here, we create a richer environment when both processes change useful insights.

### 4.4. From Design to Implementation: Feasibility Issues

This 12-week study *questions the feasibility of a randomized and longitudinal study* in a youth sports team. We aimed to understand the major difficulties within the implementation of different planning approaches into randomized groups, and how we can be able to solve them. Besides that, we intended to explore what opportunities emerge from this approach in a sports training context. Usually, 18 players represent only one group in a practice setting (i.e., one team, under one global training planification). Only under these conditions (i.e., a proper coaching staff), was it possible to provide constant feedback and have a positive interaction with players in both groups. Moreover, and according to adherence rates of players, we felt that independently of the group (experimental or contrast), there were particular cases of athletes with around 60% adherence. Here, learning improvements were difficult due to the absence of training stimuli. The idea of maintaining the pre-established plan, independent of adherences, was even more difficult than the learning adaptations.

During those 12 weeks, we attempted as much as possible to increase the number of coaches participating in practice, which did not happen all the time. Those moments (when only one coach was responsible for the eighteen players in two different groups) demanded an extra-effort from the coach and might not have created the ideal conditions for practice. The coaching team sought to fulfill this gap by creating moments before the practice for players to practice volleyball without imposed rules, and with the teams created by themselves. In this way, the sorts of players that are more motivated by being with their friends maintain those moments. Other strategy to solve this specific problem was a punctual game situation, where the experimental and contrast group played against each other but with different constraints to each team, maintaining the focus of the main goal within the practice. For example, the experimental group is focused on improving their attack on zone 3 and they might win an extra point when they score attacking on zone 3; or if the contrast group has their major focus of the week on serve and passing, they get an extra point for serving and passing well.

*The randomization* of the whole group into experimental and contrast groups was the most accurate solution to establish two different groups with the same technical and tactical baseline. However, it created some specific issues as the motivation to practice at this particular age, and in female sports, focuses especially on friendship [82,83]. In this case, we created some specific strategies to try to attenuate those side effects of group randomization. Besides the randomization to investigate issues, players from experimental and contrast groups might belong to the same team and play together in official tournaments. Therefore, players that compete together in official competitions might practice separately during the week. We tried to minimize the issue of training separately, planning the last training session before the tournament by teams (i.e., as they will play in tournament), instead of planning by groups (i.e., experimental and contrast).

## 5. Practical Implications and Future Research Directions 

As the benefits of flexible assessment and planning were promising in this investigation, we suggest that this kind of design and approach should be deeply tested in further studies. Recommendations for future studies can be seen in Figure 4. 

## 6. Limitations

During the application of the project, the majority of the limitations were related to the parallel randomization of the group and subsequent social implications, as was previously explored in the results. Additionally, on specific occasions, the main gym of the club was not available, and the group was forced to practice in a smaller space which meant that the need to train separately, due to the randomization demands, might have reduced the practice quality. Finally, having two different plans to apply would have been easier to implement with a large coaching staff.

## 7. Conclusions

We recommend the careful planning of the learning process; however, a careful and sustainable planning process should be flexible and adaptable to multiple changes over time. Plans that are too detailed ignore intra- and inter-individual variability, inherent characteristics in any learning group. This adaptability of planning should not only be individualized to learners’ necessities, but also flexible to external constraints. These adaptations over time in planning should be guided through ongoing assessments, and these can provide useful insights about learners’ evolutions and diagnostic special needs to aid planning. A flexible plan is open to receiving information from assessment and can adapt in order to respond correctly to actual learners’ needs. In the same way, the assessment tools should adapt to different constraints, issues, and contexts that the learner needs to solve over time. The assessment itself should be assessed (i.e., meta-assessment), as a result of reflection about the learning process, and for consideration of what the next steps are that coaches and teachers want their learners to undertake. Future research is recommended in this area of planning and assessment bidirectional interaction, especially during longer periods of time and in different contexts.

## Figures and Tables

**Figure 1 children-10-00029-f001:**
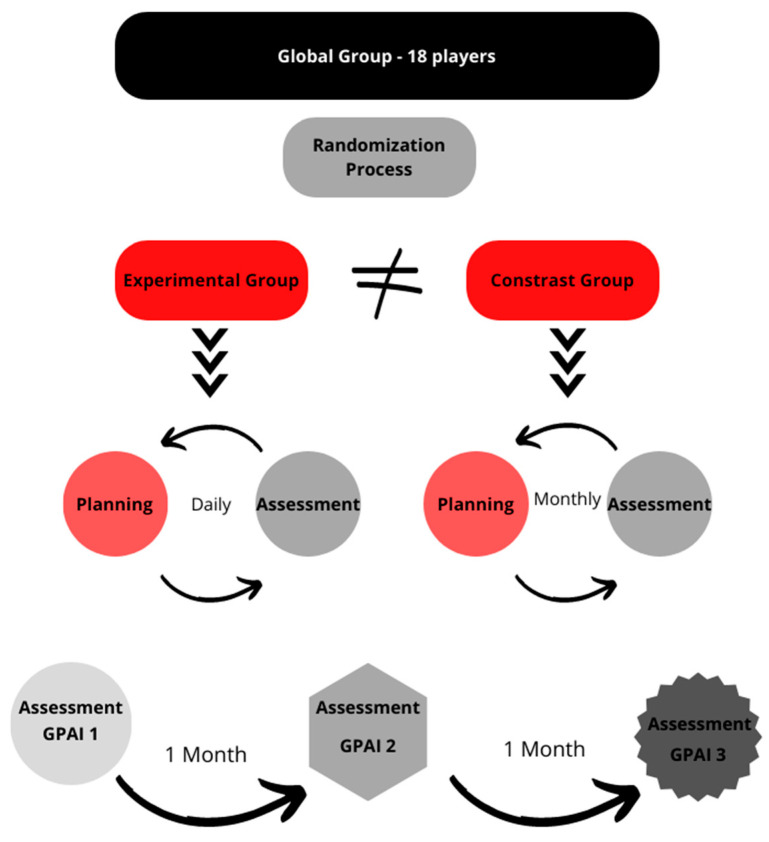
Study design.

**Figure 2 children-10-00029-f002:**
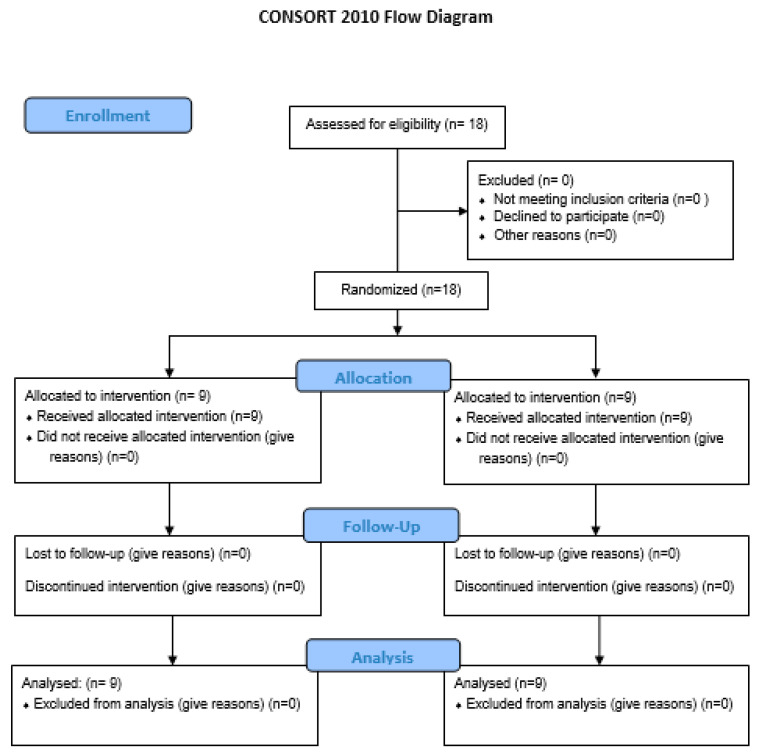
Work-flow randomization process (CONSORT).

**Figure 3 children-10-00029-f003:**
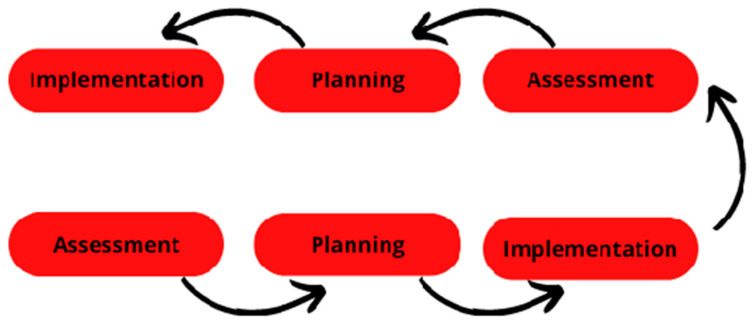
Workflow of assessment adaptation to the process.

**Figure 4 children-10-00029-f004:**
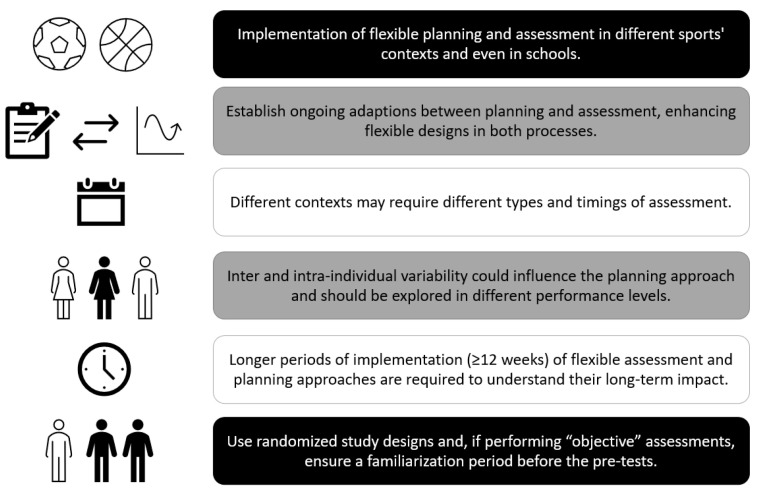
Suggestions for future research in bidirectional feedback between planning and assessment.

## Data Availability

Not applicable.

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
