# Peer review of "Flexible Training Planning Coupled with Flexible Assessment: A 12-Week Randomized Feasibility Study in a Youth Female Volleyball Team"

_children, 2022, doi:10.3390/children10010029_

Round 1

Reviewer 1 Report

Reviewer Comment

It is seen that the work you have done as an intervention study has been done in detail with a lot of effort in general. However, it is not entirely clear how your research differs from other studies in the literature, based on the topic you have chosen. Unlike other studies in the literature, you should explain more clearly what you aim to find in the introduction. The summary was too complex and incomplete as a template, the methodological information in the summary should be explained more clearly. The introduction contains a lot of general information. There is no information about the results of the studies. The hypothesis is incomplete. There is an excessive amount of information and detail throughout the entire manuscript. Detail is important and necessary, but too much destroys the clarity and fluency of the work. One of the most important things in writing an article is to present the information to be given in a concise form. If you cannot create and present the core knowledge, this article will not be a thesis. However, such resources and information are used in theses. In this sense, you should completely re-edit your manuscript. We expect you to write a more tidy manuscript with the gist of the information, with any necessary revisions. Please make the necessary arrangements carefully without skipping any revisions.

Revision

1.      The abstract section should be written in a much more systematic way. Summary; the purpose, method, findings and discussion should be explained more simply in a specific template. The number of participants and age range should be added to the method part.

2.      Page 1, line 27;  The Game Performance Assessment Instrument [1]” There is no need for reference in the summary section, delete it.

3.      Page 1, line 27; Your keywords are too general. There should have been much more specific words. There is no specific word for your research. Review the keywords again.

4.       The introduction is too long. Such a long introduction is unnecessary. In the introduction part, the essence of the subject to be explained should be explained. An introductory sentence that has turned into a long text with all the information is not suitable for the article template. Organize the introduction by giving the essence of the information and writing it more precisely.

5.      Most of the introduction consists of informative text. This type of entry is not appropriate. General information can be given in the beginning, but a flow should be created by giving similar research results in the literature and as a result, the text should be connected to the purpose of the study. Edit the introduction section template so that this is the flow.

6.      Your purpose for doing the work should be more clearly stated. Why did you decide to do this study? What was the difference from the studies in the literature? In this sense, specify the main reason for doing your work more clearly at the end of the introduction.

7.      Add the hypothesis of your study at the end of the introduction.

8.      Page 3, line 126-130;“Overall, this project presents the novelty of an evolving planning and assessment process, due to their strong, ongoing, and bidirectional links. Furthermore, to guarantee that the effects in learning were not merely due to time effects, the longitudinal nature of the intervention along with the implementation of two contrasting randomized groups ensured a greater quality of data interpretation.” After giving the purpose of the study at the end of the introduction, it is not appropriate to write an explanation similar to the general information explanation again. Add the research hypothesis only after giving the purpose statement.

9.      Are there inclusion and exclusion criteria? Add the explanation about this to the method section.

10.  Did you do a power analysis, explain how you determined the number of participants?

11.  Please provide more detailed information about the frequency and duration of the intervention in the intervention section.

12.  Page 3, line 180; You shouldn't have included the CONSORT flowchart in the results section. The flow chart is given in the result section of the systematic review. This flowchart should have been given in the participants section of the method section. Move this to the method section.

13.  The discussion was generally made with studies containing general information in the literature. This information can be used but the entire discussion is done with general information explanations. The discussion should be done by giving the results of the research in the literature and making a relationship with these results. Rearrange the entire discussion like this. Mention studies that include much more research results.

14.  What are the limitations of the research?

Author Response

Dear reviewer,

Thank you for your detailed analysis and criticisms. We have attempted to address all your points. However, we felt that a few of those points would not fit this manuscript's goals. Attached is a Word file with a point by point response, where we address everything in detail. For your convenience, the revised manuscript has all the changes marked with the Track Changes function of MS Word. 

Kind regards,

Manuel Loureiro   

Reviewer 2 Report

Dear authors

You described a well-defined research design, aiming to test the feasibility of a flexible planning coupled with a flexible 124 assessment process, launching an ongoing, bidirectional feedback between plan and assessment in a young female volleyball team. The quality of the study elaboration and results are clearly described in the text. With few exceptions where I gave my observations. No further remarks. The reviewers are in the pdf document.

Best Regards;

Author Response

Dear reviewer,

Thank you for your comments. We have attempted to address all your points.  Attached is a Word file with a point by point response, where we address everything in detail. For your convenience, the revised manuscript has all the changes marked with the Track Changes function of MS Word. 

Kind regards,

Manuel Loureiro   

Round 2

Reviewer 1 Report

Reviewer Comment

It is seen that the revisions that I have suggested in general have been made in the manuscript. However, there are a few places that need to be regulated. There are resources that you have deleted and added in the manuscript. These should be arranged in order. It is also seen that some sources are very old. Remove these sources from the manuscript and include more up-to-date sources instead.

Author Response

Dear reviewer,

Thank you for your recommendations. We have carefully looked to all sources and updated the older ones. In this new version we did not use Track Changes tool of Word, in order to update all the references in the correct order. Instead, the changed references are highlighted in yellow.

Kind regards,

Manel Loureiro